# Influence of Carob Flour and Carob Bean Gum on Rheological Properties of Cocoa and Carob Pastry Fillings

**DOI:** 10.3390/foods8020066

**Published:** 2019-02-12

**Authors:** Maja Benković, Tomislav Bosiljkov, Amela Semić, Damir Ježek, Siniša Srečec

**Affiliations:** 1Faculty of Food Technology and Biotechnology, University of Zagreb, Zagreb 10000, Croatia; mbenkovic@pbf.hr (M.B.); tbosilj@pbf.hr (T.B.); djezek@pbf.hr (D.J.); 2Faculty of Healthcare, University of Vitez, Travnik 72270, Bosnia and Herzegovina; amelas_vi@yahoo.co.uk; 3Križevci College of Agriculture, Križevci 48260, Croatia

**Keywords:** cocoa, carob, locust bean gum, modelling, pastry fillings

## Abstract

The aim of this study was to develop a new cocoa and carob based pastry filling and explore the influences of carob flour and carob gum on the rheological and textural properties, specifically (i) the effect of increasing the amount of carob flour and (ii) the effect of carob bean gum naturally present in the carob flour with seeds versus the commercially available carob bean gum. All samples analyzed in this study exhibited shear thinning behavior. The texture analysis revealed a significant (*p* < 0.01) increase in consistency and firmness in samples with higher amounts of carob flour added, while higher temperatures significantly (*p* < 0.01) decreased adhesiveness. When comparing naturally occurring and commercially available LBG (locust bean gum), it was concluded that lower concentrations (up to 0.45% *w*/*w*) of naturally occurring LBG work just as well at the same concentrations of commercially available LBG, but this effect cannot be confirmed for higher LBG concentrations, nor for rheological properties determined at higher temperatures (80 °C).

## 1. Introduction

Pastry fillings represent a combination of different ingredients, such as sugar, cocoa, nuts, fruits, stabilizers and emulsifiers added to the final product. Nowadays, a large portion of pastry fillings is commercially available and sold in plastic packaging of 1 kg and up, ready to use in the production process. The rheological behavior and flow properties of pastry fillings have significant roles in the food industry as they govern the product development, design and evaluation of the process equipment [1]. Pastry fillings are usually regarded as non-Newtonian fluids, mostly exhibiting pseudo plastic behavior, with a decrease in the viscosity as the shear rate increases. Due to changing consumer preferences and consumer demands for the development of new, functional foods, the food industry nowadays is faced with the challenge of developing products with a lower sugar content, lower fat content and a higher amount of antioxidants while maintaining good rheological and textural properties. In the case of pastry filling, new products with different flavor combinations emerge every day. 

Cocoa-based fillings are one of the most popular types in the market nowadays. In most cases, they comprise cocoa powder, milk powder, fat, sugar and additives for texture and stability improvement. Another type of pastry filling that is also present on the market but is often neglected and outrun by the more popular cocoa filling is carob filling. Carob-based fillings represent a cheaper alternative with numerous beneficial properties owing to their high content of polyphenols [2,3]. Besides bioactive compounds, previous research has documented the presence of carbohydrates (76%), proteins (6%), fat (2%) and minerals in carob flour [4]. When roasted, carob flour has a similar taste and sensory properties to chocolate, but consumers still prefer cocoa powder-based treats to those made only from carob. However, this problem could be overcome by developing new pastry filling recipes in which carob would be combined with cocoa powder and therefore gain better consumer acceptance. In comparison to cocoa, carob contains less bioactive compounds, but is richer in dietary fiber, which benefits digestion, and it has a higher carbohydrate content, indicating that, in the production process, sugar addition to the mixture can be diminished when adding carob. The high carbohydrate content of carob flour could prove beneficial for flavor acceptance, but in the case of rheological properties, sugars can affect texture by retarding the flow as well as by modifying the gelling characteristics [5]. 

Carob is also used as a raw material for the production of locust bean gum (LBG). LBG, also known as carob gum, is a natural hydrocolloid, which has been used in numerous industrial applications because of its ability to enhance viscosity at relatively low concentrations (≤0.02%) to stabilize emulsions and as a fat replacement. In the food industry, LBG alone or in combination with other hydrocolloids is used in beverages, bakery products, noodles, dairy products and edible coatings [6]. The production process of LBG gum usually requires chemical or thermo-mechanical treatment of seeds [7] and the use of many chemicals, solvents and energy, so the possibility for its direct use as a part of carob flour produced without removing the seeds has to be further explored.

Taking everything mentioned into account, the aim of this study was to develop a new cocoa and carob-based pastry filling and explore the influences of carob flour and carob gum on the rheological and textural properties of the fillings. Two factors were varied in order to get an insight into the rheological properties: (i) the effect of increasing the amount of carob powder and (ii) the effect of carob bean gum naturally present in the carob flour with seeds versus commercially available carob bean gum.

## 2. Materials and Methods 

### 2.1. Materials

Cocoa powder (10–12% fat content) and crystal sugar were supplied by Kraš d.d. (Zagreb, Croatia). Whole milk powder (Dukat, Zagreb, Croatia), vegetable fat (Zvijezda d.d., Zagreb, Croatia) and carob bean gum (Biovegan, Bonefeld, Germany) were purchased from a local store. Carob flour was supplied by a local manufacturer from the island of Drvenik Mali in Croatia, and two types of carob flour were used in pastry filling production: carob flour produced from dekibbled carob pods and carob flour produced from whole carob pods (including seeds).

### 2.2. Methods

#### 2.2.1. Pastry Filling Production

Pastry fillings were manufactured based on the following design: 3 sets of samples of which the first set (A) was considered a control set containing carob flour without seeds, the second set (B) contained carob flour with naturally present seeds and the third set (C) contained carob flour without seeds and commercially available carob bean gum. All samples comprised the same basic ingredients: 45 g sucrose, 30 g vegetable fat and 10 g whole milk powder, while only the amounts of cocoa, carob flour and commercially available locust bean gum were varied. The exact list and compositions of the samples are shown in Table 1.

The amount of LBG added to set C of the samples was calculated based on the morphological characteristics of carob pods from which the flour was produced (number of seeds, seed weight, portion of the endosperm), as published previously by [8]. Samples were mixed in a kitchen blender (Philips, Amsterdam, The Netherlands) using a flat beater equipped with a flexible edge to allow successful separation of the materials from the edges of the mixing bowl. The samples were mixed for 7 min to obtain a homogenous blend and were transferred to plastic pouches until further analysis. Prior to measurements, samples were conditioned at 45, 60 and 80 °C in an IKA HBR digital water bath (IKA Werke, Staufen, Germany) in order to show the effect of temperature on the rheological and textural properties of the fillings.

#### 2.2.2. Rheological Analysis

Rheological analysis was performed using FungiLab Alpha L rotational viscometer (Barcelona, Spain) equipped with an L4 spindle (length 31.2 mm, diameter 3.3 mm). Spindle selection was carried out based on a preliminary experiment on 4 different spindles (data not shown), based on the “trial and error method”, as suggested by the manufacturer of the viscosimeter. The same spindle was selected for all samples in order for the results to be comparable, keeping in mind the acceptable viscosity ranges determined by the manufacturer of the instrument listed in the user manual. Samples were tempered in a water bath to ensure the right temperature during measurements (45, 60 and 80 °C) and left to settle for 10 min before performing the measurements. Measurements were done at 18 pre-defined rotational speeds: 0.3, 0.5, 0.6, 1, 1.5, 2, 2.5, 3, 4, 5, 6, 10, 12, 20, 30, 50, 60 and 100 rpm, at which the apparent viscosity values were determined. Measurements were done in triplicate.

#### 2.2.3. Mathematical Modelling

Experimental data was used for mathematical modelling in the following manner: First, the apparent viscosity data were used to calculate the shear stress values at a given shear rate. The obtained calculated shear stress values were then fitted to two models: Ostwald de Waele (Equation (1)) and Herschel–Bulkley (Equation (2)).
(1)$$\tau =k\times {\dot{y}}^{n}$$
(2)$$\tau =\tau_{y}+k\times {\dot{y}}^{n}$$ where *τ* is the shear stress (in Pa), *τ_y_* is the yield stress (in Pa), *k* is the consistency index (in Pas^n^), *n* is the flow behavior index and *ẏ* is the shear rate (in s^−1^). Model fitting was performed using Statistica v. 10 software (StatSoft, Tulsa, OK, USA). In order to remove any measurement noise from the experimental data, prior to mathematical modelling, raw data was pre-processed by the Box-Whisker method, and the points marked as outliers were not used in further mathematical modelling. Experimental data was then fitted to the models using user-defined non-linear regression by the Levenberg–Marquardt algorithm. The performance of the developed models was evaluated according to the following statistical parameters: *R*^2^, root mean square error (RMSE) and mean percentage error (MPE), which allowed the detection of differences between the experimental data and the model estimates [9]. The RMSE was calculated based on Equation (3):(3)$$RMSE={\left[\frac{1}{N}\overset{N}{\underset{i=1}{\sum }}{\left(\tau_{exp,i}-\tau_{pred,i}\right)}^{2}\right]}^{\frac{1}{2}}$$ where *N* is the number of data points, *τ_exp,i_* is the experimentally-determined shear stress and *τ_pred_*_,*i*_ is the model-derived shear stress. Higher values of *R*^2^ and lower RMSE values indicate better adequacy of the model for data approximation. However, since literature data [10] states that, in some cases where model sensitivity was evaluated only by RMSE, data evaluation is not critical, because variations of the same model have similar error distributions, so the mean percentage error was also used (Equation (4)):(4)$$MPE=\frac{1}{N}\sum_{i=1}^{N}[\frac{\tau_{exp,i}-\tau_{pred,i}}{\tau_{exp,i}}]\times 100.$$

#### 2.2.4. Texture Analysis

The texture analysis was performed using a TA.HD Plus Texture analyzer (StableMicro Systems, Godalming, UK). Samples were analyzed by a back extrusion test using a 30 kg load cell and a 45 mm diameter back extrusion cell. Test settings were as follows: test speed, 1 mm/s; distance, 10 mm and trigger force, 10 g. Prior to each measurement, samples were conditioned at appropriate temperatures (45, 60 and 80 °C) using an IKA HBR 4 digital water bath (IKA Werke, Staufen, Germany). The maximum extrusion force (Fmax, in g) and the area under the curve (A, in g·s) were obtained as indices of firmness and consistency, while the area under the negative part of the curve was used as the index of viscosity (adhesiveness) (g·s).

#### 2.2.5. Data Analysis and Statistics

Data analysis was performed using Statistica v.10 software (StatSoft, Tulsa, USA), which was used to conduct the Least Significant Difference test. Samples were treated as dependent samples and the differences were accepted as significant at *p* < 0.01. To detect samples with similar patterns of textural and viscoelastic properties, Principal Component Analysis (PCA) was applied using the trial version of XLStat software (Addinsoft SARL, New York, NY, USA) coupled with Microsoft Office Excel (Microsoft, Redmond, DC, USA).

## 3. Results and Discussion

### 3.1. Flow Behavior

Flow curves for the prepared pastry fillings are shown in Figure 1.

A detailed analysis of the viscosity of multicomponent (heterogeneous) mixtures was carried out by comparison with the reference sample as a function of the observed temperature interval. The aim of this analysis was to obtain a consistent value for viscosity (interpolation) where the system components interact in a manner that forms adsorbed layers of smaller particles on the surface of larger ones [11]. In this study, A, B0 and C0 were considered reference samples.

Based on the data shown in Figure 1, comparing the samples of group A, B and C, slight variations in the change in apparent viscosity at all shear rates were visible. The most pronounced changes, which were a result of the complexity of the sample composition of all groups, were visible at lower shear velocities in the interval of 0.3–10 min^−1^. More pronounced oscillations were observed at temperatures of 60 °C where the apparent viscosity of all samples of groups A, B and C (Figure 1B,E,G) relative to the reference samples (A0, B0 and C0) was reduced. The reduction in viscosity was most pronounced at lower shear rates, but the proportional decrease in the observed value was visible up to a maximum rate of 100 min^−1^. At temperatures of 80 °C (Figure 1C,F,I), in contrary to the expected influence of temperature on apparent viscosity, the viscosity of the sample sets A, B and C was similar to the values measured at 45 °C (Figure 1A,D,G), which was a result of mixture disintegration that was visible with the naked eye at higher testing temperatures (80 °C). In general, the apparent viscosities for all fillings decreased with an increasing shear rate, which is an indication of the shear thinning properties. According to [12], this behavior can be explained as a consequence of aggregate formation due to high molecular weights—at low shear rates, aggregates can remain strongly bound together, but they can also be broken at high shear rates. Furthermore, according to [13], at lower shear rates, disruption of molecular entanglements is balanced by the formation of new entanglements so that the viscosity can be kept constant. At higher shear rates, disruption of the entanglements becomes dominant and the molecules align in the direction of flow which leads to a decrease in viscosity. Furthermore, we must emphasize that for samples containing larger percentages of carob flour, the occurrence of slippage cannot be excluded due to the presence of larger and firmer carob flour particles, which can cause an increase in sample roughness and its attrition to the spindle surface.

### 3.2. Model Fitting

Experimental data obtained by the viscosimeter were further subjected to mathematical modeling. The data were fitted to the two most commonly used models describing the behavior of non-Newtonian fluids: Ostwald (also known as the power law model) and Herschel–Bulkley. 

#### 3.2.1. The Ostwald de Waele Model

The Ostwald model describes the behavior of non-Newtonian fluids with the introduction of two model parameters: *k* and *n*. The consistency coefficient, or flow consistency index (*k*), gives an idea of the viscosity of the fluid, with the units Pas^n^ which represent shear stress at a shear rate of 1.0 s^−1^ and the flow behavior index (*n*), a dimensionless parameter that gives fluid flow behavior information and reflects the closeness to Newtonian flow [14,15]. While the values of *k* can have different values and ranges depending on the sample tested, *n* values are used for classification of fluids: if *n* ranges from 0 to 1, the fluid is categorized as pseudoplastic; if *n* is 1, the Ostwald de Waele equation turns into the Newton viscosity law, and the fluid is Newtonian, and if the value exceeds 1, the fluid is considered dilatant. However, to be able to compare the values of *k* for different fluids, they should have similar flow behavior indexes [14]. As shown in Table 2, *n* values for all samples at all temperatures were lower than 1, indicating a pseudoplastic behavior of the pastry fillings. For pastry fillings at 45 °C, flow behavior indices ranged from *n* = 0.74 detected for the “control” samples, A0, B0 and C0, without gum or carob flour addition to a maximal *n* = 0.82 for sample B3 made with carob flour which contained seeds. In the case of set A, all samples were made with seedless carob flour, and the increased amount of carob flour in the fillings led to a decrease in the *n* value, indicating a more pronounced shear thinning behavior with a decrease in the apparent viscosity at higher shear rates. In the case of set B, the *n* value for the B1, B2 and B3 samples was also higher in comparison to that of the control sample B0, but no apparent rise or fall trend was detected in relation to the amount of carob flour contained in the sample. Set C, which was made with addition of commercially available locust bean gum, exhibited a slight rise in *n* with the rise of carob flour and locust bean gum in the sample (*n* = 0.76 for C1, *n* = 0.76 for C2 and *n* = 0.81 for C3, respectively) which was an indication that the shear thinning behavior was halted at higher carob flour and LBG contents. When comparing all three sets of samples at the temperature of 45 °C, the flow consistency index (*k*) was the highest for set B, followed by sets C and A. According to this data, B samples exhibited the highest viscosity at temperature of 45 °C. Determination coefficients for the Ostwald models for 45 °C indicated an excellent fit of the experimental data to the proposed model (*R*^2^ ranged from 0.9969 for A3 to 0.9999 for B2, respectively), and this was also seen when considering low root mean square error values: the lowest RMSE = 0.03 Pa (B2) and the highest RMSE = 0.13 Pa (A0, B0 and C0). However, when considering the MPE as a means for evaluating modelling performance and accuracy, literature data state that the models are considered reliable when the MPE values are lower than 10% [8]. In this case, MPE values lower than 10% were achieved only for samples A1, A3, B2 and C1, indicating that the modelling performance cannot be considered reliable, despite the high *R*^2^ and low RMSE values. 

At a temperature of 60 °C, the flow behavior indices of all sample sets were also lower than 1, meaning that the shear thinning behavior continued at higher temperatures. Among sample sets, a decrease of *n* values was detected with an increase of carob flour and LBG in the fillings. The same was detected for the *k* values, which is an indication that the viscosity of the fillings decreases with higher levels of carob flour present in the fillings. Values of the determination coefficient were all higher than *R*^2^ = 0.9000, demonstrating an excellent fit of the proposed model to the experimental data. It is important to emphasize that, in comparison to the *R*^2^ and RMSE values determined for the lower temperature of 45 °C, in most cases, the *R*^2^ values were lower and the RMSE values were higher at 60 °C. This is an indication that, although strong links were found between observed and model predicted data, bonds tended to weaken when the viscosities were determined at higher temperatures. This was also confirmed by the RMSE and *R*^2^ values calculated for experimental data determined at 80 °C, in which case some of the *R*^2^ were lower than 0.9000 (*R*^2^ = 0.8669 for A2 and *R*^2^ = 0.8319 for C3), and the RMSE values were as high as RMSE = 0.51 (A0, B0 and C0). When considering the modelling performance based on MPE values, it can be seen that reliable model values (MPE < 10%) were achieved at 60 °C for samples with lower amounts of carob flour, while the Ostwald model cannot be considered reliable in the case of samples A3, B3 and C3, which exhibited high MPE values (19.13, 20.97 and 24.29%, respectively).

At 80 °C, the consistency index (*k*) again showed a decrease as the amount of carob flour in the fillings increased, with an exception of set B. This exception can be attributed either to experimental error or a difference in the model fitting accuracy. The highest k value was detected for the control samples A0, B0 and C0. A drop in flow index was also detected with higher carob flour contents. As stated earlier, *R*^2^ values determined for the temperature of 80 °C were lower in comparison to 45 and 60 °C, and the RMSE value was as high as RMSE = 0.51, indicating a lower accuracy of the Ostwald model for higher temperatures. This was also confirmed by the MPE values which were much higher than 10% for the 80 °C analyses, indicating that the Ostwald model cannot be applied for viscosity data approximation of cocoa and carob pastry fillings at 80 °C.

According to the literature data, the influence of temperature on the viscosity of the food products can, in most cases, be described as a decrease in viscosity at higher temperatures, e.g., Reference [16] investigated the influence of temperature on the viscosity of, among others, mango pulp and apple syrup and concluded that higher temperatures decrease the values of the consistency coefficient and the flow behavior index. Reference [17] described the influence of temperature on the viscosity of tapioca meal. Higher temperatures reduced the viscosity, but the basic rheological behavior of the meals remained pseudoplastic (*n* < 1) at all analyzed temperatures.

In this research, the influence of temperature on consistency coefficients (*k*) did not yield obvious rise or fall trends, except in samples A1, B1, B2, C1, C2 and C3, indicating that the influence of temperature on viscosity is easier to detect in samples with the addition of naturally occurring or commercially available locust bean gum. In those samples, viscosity decreased with an increase in temperature. The decrease in viscosity can be attributed to the increase in intermolecular distances, because of the thermal expansion caused by the increase in temperature [16]. Reference [18] stated that, when heating the LBG solution above 80 °C, oxidative-reductive depolymerisation of galactomannan chain and reduction in viscosity of the final solution may be observed, which is also an explanation for why the mathematical models failed to perform at 80 °C (high MPE). Among other samples (A0, A2, A3, B3), structure collapse and breakage of sample segments was visually present in the measuring cylinder during measurement, which could explain the lack of a universal temperature dependence trend in those samples. The flow behavior index remained lower than 1 at all temperatures for all samples, demonstrating that, although a slight decrease of n was visible at higher temperatures, flow behavior can be described as pseudoplastic throughout the whole testing range.

#### 3.2.2. Herschel–Bulkley Model

To see whether the cocoa and carob pastry fillings require initial stress to begin to flow, fitting of the experimental data to the Herschel–Bulkley model was also performed. Besides the flow behavior index *n* and the consistency coefficient *k*, the model introduces the parameter *τ_y_* which represents the yield stress, which quantifies the amount of stress that the fluid may withhold before it starts to flow. Estimated parameters *k*, *n* and *τ_y_* as well as the *R*^2^, MPE and RMSE values for the Herschel–Bulkley equation are shown in Table 3.

Values of the estimated *τ_y_* parameter for a temperature of 45 °C shown in Table 3 range from *τ_y_* = 0.02 Pa (C1) to 0.19 Pa (C2). It can be noticed that the model fitting resulted in negative *τ_y_* values for samples A0, B0, B3 and C0. This can be interpreted in two ways: First, these yield stress values can be considered as *τ_y_* = 0 Pa, and the flow curves in that case are approximated by the Ostwald model. It is also worth mentioning that none of the model-estimated negative parameter values were considered significant at *p* < 0.05. The other way to interpret the data is connected to the procedure of determination of *τ_y_*. Namely, [15] emphasized that *τ_y_* should first be experimentally determined, and the rest of the parameters (*k* and *n*) then estimated by means of linear regression. In this study, *τ_y_* was not determined experimentally, but only estimated by means of non-linear regression. According to [15], estimated parameters can only be used if experimental determination of *τ_y_* is not available, and nonlinear regression provides values that are the best in a least squares sense and may not reflect the true nature of the test sample. The negative values of *τ_y_* obviously do not present logical findings, based on which it can be concluded that, although the R^2^ values would propose an excellent fit of the Herschel–Bulkley model to experimental data, the interpretation of such *τ_y_* data has to be made with caution. However, when considering the MPE as a statistical means of evaluation of modelling data, all values for 45 °C were lower than 10%, which is an indication that the Herschel–Bulkley model can be applied successfully. Similar as with the Ostwald model, *k* values ranged from *k* = 5.08 Pas^n^ (B1) to *k* = 2.04 Pas^n^ (A3) and exhibited the tendency to decrease with larger amounts of carob flour present in the filling. The flow behavior index ranged from *n* = 0.71 (A0, B0, C0) to *n* = 0.91 (C3), and no obvious trend of change was detected connected to the amount of carob flour present in the sample. The range of *n* values demonstrated that all mixtures have pseudoplastic behavior at 45 °C.

At 60 °C, *τ_y_* values increased slightly in comparison to data at 45 °C, indicating that, at higher temperatures, the filling starts to flow after the initial stress value has been reached. The yield stress values increased with higher contents of carob flour in the sample. In all sample sets, viscosity decreased with an increase of carob flour in the samples. The decrease in viscosity could be explained by the particle size of the carob flour added to the fillings. Namely, cocoa powder used in this study had a median diameter of d(0.5) = 13.32 µm, while carob flours had median diameters of d(0.5) = 126.01 µm (for the flour without seeds) and d(0.5) = 145.81 µm (for the flour with seeds). As previous research stated, particles which are smaller in diameter have a larger specific surface area and are therefore more prone to the creation of inter particle bonds [19]. In practice, they tend to form larger clusters or stick to the surfaces of larger particles and are more interconnected. In this case, smaller cocoa particles created a filling with more inter particle bonds and thus had higher viscosity. As carob flour was introduced to the mixture, larger particles with lower specific surface area did not form as many interparticle bonds, and thus there was a reduction in filling viscosity. A similar effect of decreasing the particle size on viscosity was also documented by [20], who concluded that in high viscosity fluids, increasing the particle size leads to a decrease in viscosity. Values of the flow index determined at 60 °C ranged from *n* = 0.69 (A0, B0) to *n* = 0.95 (A3), leading to the conclusion that the flow behavior of the fillings at 60 °C is also pseudo plastic. Model adequacy estimation parameters (R^2^, RMSE and MPE) indicated that the Herschel–Bulkley model can be successfully used to describe the flow behavior of cocoa and carob pastry fillings at 60 °C. However, the situation changes at 80 °C. Namely, at 80 °C, during visual inspection of the sample in the measuring cylinder, structure tears and separation of the segments of the fillings were visible. Because of that, fitted parameters for the Herschel–Bulkley model differed significantly (*p* < 0.05) from the values determined for lower temperatures, e.g., the control samples A0, B0 and C0 exhibited an *n* value of *n* = 1.09, which changes the samples’ flow behavior from pseudo plastic to dilatant. The same was noticed for samples A2, C1 and C3. Determination coefficients for model parameters at 80 °C were lower in comparison to those at 45 and 60 °C, the RMSE values much higher, and the MPE values crossed the threshold of 10%, which is an indication of a lower fit to experimental data and lower modelling accuracy and reliability.

When comparing the fitted parameter values, R^2^, RMSE and MPE values for both the Ostwald and Herschel–Bulkley models, it can be seen that, despite high R^2^ and low RMSE values, the MPE values indicate that model approximation is not adequate. This was especially pronounced for the Ostwald model, even at lower temperatures (45 and 60 °C) and was especially visible at 80 °C for both models, where mathematical modelling cannot be applied and considered reliable. The approximation of experimental data by the Herschel–Bulkley model was successful for temperatures of 45 and 60 °C, but not for 80 °C. A flow behavior description of cocoa and carob pastry fillings at higher temperatures by other flow behavior models that were not included in this study must be included in future research.

#### 3.2.3. Naturally Occurring LBG vs. Commercial LBG

As stated previously, sample sets B and C were produced with carob flour with naturally present carob seeds (B) and with the addition of commercially available LBG (C). For sample set B, the gum content was calculated based on previously published data on the morphology of carob pods used for flour production [8], and the corresponding amount of commercial LBG was added to the sample set C. The gum contents were as follows: 0% (B0 and C0), 0.15% (B1 and C1), 0.30% (B2 and C2) and 0.45% (B3 and C3). When discussing the LBG content of the samples and taking into account the reasons for LBG gum use in food applications, the decreasing *k* values with an increasing content of LBG gum would represent a questionable finding. However, as argued before, besides the increase in the LBG gum content in sample sets B and C (B1 > B2 > B3 and C1 > C2 > C3, respectively), the content of carob flour increased as well. In this case, the presence of larger particles in the fillings had a prevailing influence over the LBG content and caused the viscosity decrease. According to [16], the thickening capacity of LBG is dependent on numerous factors, including the particle size.

When comparing the B and C samples with the corresponding concentration of LBG, it was noticed that, at 45 °C, *k* values were higher for sample set C which contained commercially available LBG (Table 3). The flow behavior index of both B and C sample sets was lower than 1, indicating that the presence of LBG did not influence the type of flow behavior exhibited by the fillings. Slight differences were also found between the *n* values of the sample sets, but, although detected, these differences were not significant (*p* < 0.05). However, it is important to emphasize that, especially at higher temperatures, the influence of naturally occurring and commercially available LBG cannot be compared due to the high MPE values which indicated a poor fit of model-derived parameters to the experimental data.

### 3.3. Textural Properties

The textural properties of cocoa and carob based fillings were analyzed by means of a back extrusion test. The obtained results for firmness and consistency are shown in Figure 2.

As seen in Figure 2A, the consistency of the samples increased with an increase in the carob flour content. Namely, carob flour is high in total sugar content, ranging from 48 to 56%, which mainly comprises sucrose, glucose and fructose [21]. According to [1], the sugar content affects properties such as gelatinization, retrogradation and stalling, and their study on the effect of sugar addition on raspberry cream filling concluded that the addition of all examined sugars (sucrose, fructose and trehalose) increased the cream filling consistency. The same was observed in this study—an increased content of carob flour, and hence, an increased content of sugars, caused a rise in the consistency of the fillings. Consistency, if often used by the manufacturers, is a check of the quality of the finished product, and it is generally considered that higher sample consistency is a favorable property. However, consistency has to be kept in a range where it is sensory acceptable to the consumers, e.g., [22] studied sensory, viscoelastic and textural properties of newly developed fruit fillings and reported that, in some cases, fruit fillings were described as “too consistent”, which lowered the sensory scores. In this case, it can be seen that samples with the highest content of carob flour have significantly higher consistency values, which puts them in a risk of being rejected by the consumers as being too consistent. Therefore, complete substitution of cocoa with carob in this case cannot be recommended.

Firmness values of analyzed samples (Figure 2B) also increased with an increase of the amount of carob flour in the filling, which is not a favorable property, since fillings with higher firmness tend to haul the consumers away from the product.

By analyzing the values of consistency and firmness at different temperatures, no universal trend was detected on how these values change with temperature changes.

Adhesiveness data are shown in Figure 2C. Generally speaking, the highest adhesiveness values were detected for samples analyzed at 45 °C, with a decrease in cohesiveness detected at higher temperatures. The addition of carob flour appeared to have the most pronounced effect on adhesiveness for samples analyzed at 80 °C, which was especially visible in samples containing commercially accessible LBG. This effect was not detected for samples with naturally occurring LBG (sample B). Sample set B showed an increase in adhesiveness with an increase in carob flour addition, while no evident trend on the temperature dependency was detected for sample set A. These results are a confirmation of the beneficial influences of galactomannans from LBG on the textural properties of pastry filling, with an emphasis on better controllability of textural properties when commercial LBG is used. This can be explained by the presence of impurities in naturally occurring LBG (especially husk and germ parts), since those impurities are known to have an effect on the pasting and emulsifying behavior of LBG [18].

### 3.4. Principal Component Analysis

Data for viscosity determined by a viscosimeter at all shear rates as well as the data from the texture analysis were used for principal component analysis. The results are shown in Figure 3.

First, two factors determined by the PCA analysis accounted for 91.94% of the variability between the samples at 45 °C (Figure 3A), 82.53% at 60 °C (Figure 3B) and 83.06% at 80 °C (Figure 3C). At 45 °C, there was an evident grouping of the texture variables (consistency and firmness) on the left side of the factor plane and the viscosimeter data on the right side of the plane. Samples A3, B3 and C3 were also grouped on the left side of the plane, which means that they were separated from the rest due to their high consistency and firmness values. Samples with lower amounts of carob flour were all grouped on the right side of the factor plane and were separated based on their viscosity values at higher shear rates (5–100 min^−1^) (Figure 2A).

At a temperature of 60 °C, grouping of the samples was also exhibited, but less variability was explained by the factors in comparison to at 45 °C. Consistency and firmness were again grouped on the left side of the plane, but this time they were closer to the vertical axis, while the viscosimeter data remained grouped on the right side of the plane (Figure 2B). A clear distinction between the control samples (A0, B0 and C0) was visible in the fourth quadrant as well as for samples A3, B3 and C3 in the second quadrant, once again separated based on their consistency and firmness. Grouping of the samples with lower contents of carob flour was visible in the third quadrant, but it could not be connected to the viscosity and textural properties determined in the study. Presumably, the separation could be based on the composition of the fillings.

Finally, the PCA analysis for samples at 80 °C (Figure 3C) showed a change in positioning of the consistency and firmness data, which were situated in the fourth quadrant. Once again, samples with the highest content of carob flour (A3 and C3, with an exception of B3), were separated based on their textural data. Viscosimeter data were also visible on the right side of the plane, and a group of samples (B1, B2, B3 and C2) were placed in the third quadrant and presumably separated only based on their compositions.

Furthermore, when comparing the grouping of variables at different temperatures, it was noticed that the viscosimeter data exhibited a shift from the first quadrant grouping to the second quadrant, which was a confirmation of the influence of temperature on the viscoelastic properties of the fillings. Secondly, the data for 80 °C exhibited the opposite grouping trend of the variables in comparison to that of the 45 and 60 °C data. Namely, the data for the lowest shear rate was situated in the fourth quadrant, moving upwards toward the third quadrant where the highest shear rate was located. Data for 45 and 60 °C had the highest shear rates in the fourth quadrant and the lowest in the second. This is an indication of the flow behavior change at 80 °C, which was also detected by model approximation, where *n* values for some samples at 80 °C were higher than 1, which puts them in the dilatant behavior mode.

## 4. Conclusions

The effect of naturally occurring LBG in comparison to the commercially available LBG and the presence of carob flour on the viscosity and textural properties of cocoa and carob pastry fillings was studied. All fillings analyzed in this study exhibited shear thinning behavior, which was a result of the complexity and multicomponentiality of the produced mixtures. A texture analysis revealed that a larger content of carob flour in the samples led to higher consistency and firmness values, with no evident trend of temperature dependence. When comparing naturally occurring and commercially available LBG administered at a lower concentration range, the naturally occurring LBG had a similar effect on viscosity and texture. However, this effect was not confirmed for higher concentrations of LBG, nor for its effect on rheological properties measured at higher temperatures (80 °C).

## Figures and Tables

**Figure 1 foods-08-00066-f001:**
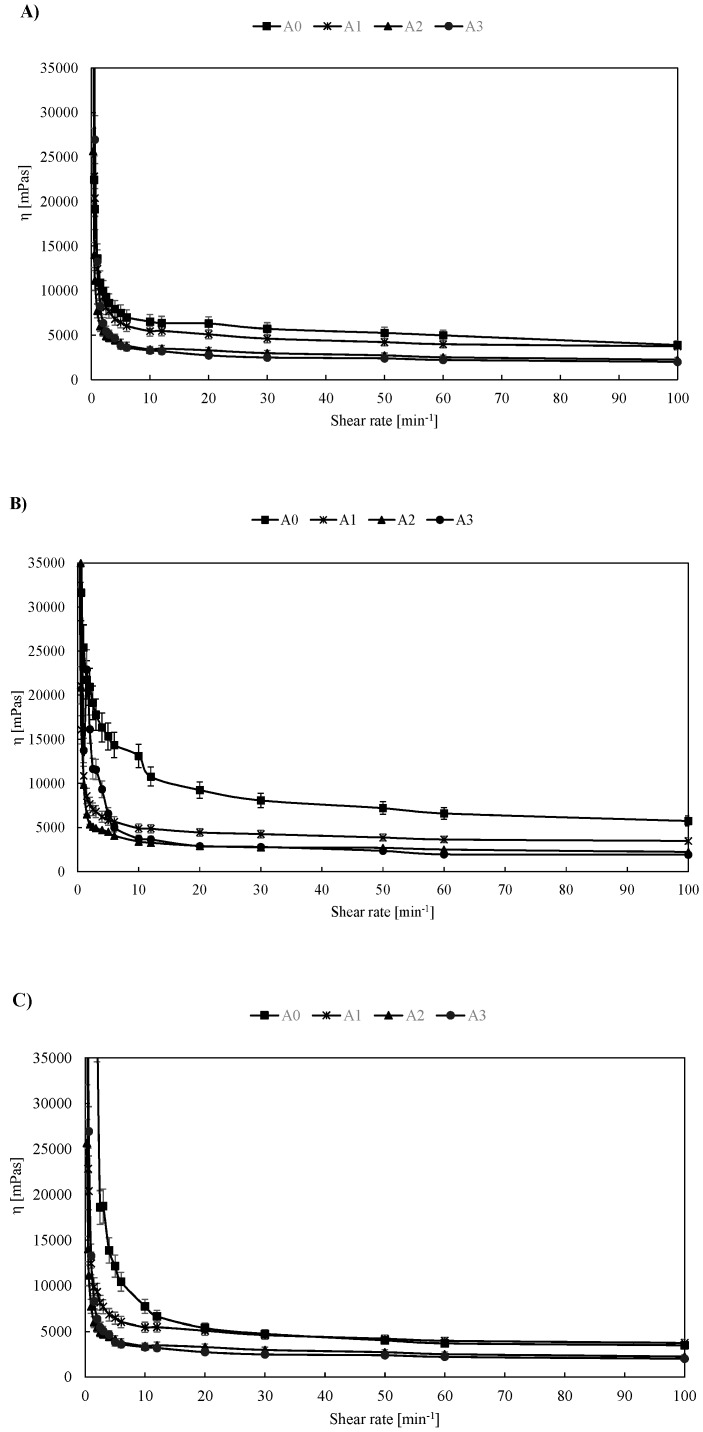
Flow curves for (**A**) sample set A at 45 °C; (**B**) sample set A at 60 °C; (**C**) sample set A at 80 °C, (**D**) sample set B for 45 °C; (**E**) sample set B for 60 °C, (**F**) sample set B at 80 °C, (**G**) sample set C at 45 °C, (**H**) sample set C at 60 °C and (**I**) sample set C at 80 °C. Results are expressed as mean (*n* = 3) ± SD (standard deviation). To allow easier comparison values, viscosity at low shear rates is omitted from the charts, and the shear rate is expressed as min^−1^.

**Figure 2 foods-08-00066-f002:**
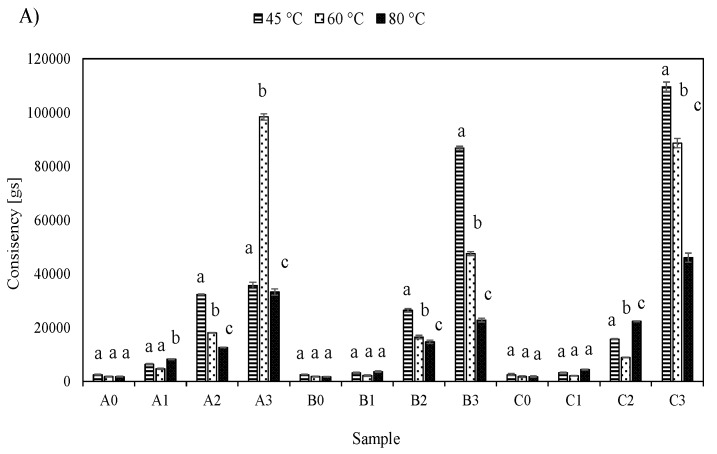
Textural properties of cocoa and carob-based pastry fillings: (**A**) consistency, (**B**) firmness and (**C**) adhesiveness (results are expressed as mean values of 3 measurements ± SD). Different letters above the column for the same sample analyzed at different temperatures represent significant differences at *p* < 0.01.

**Figure 3 foods-08-00066-f003:**
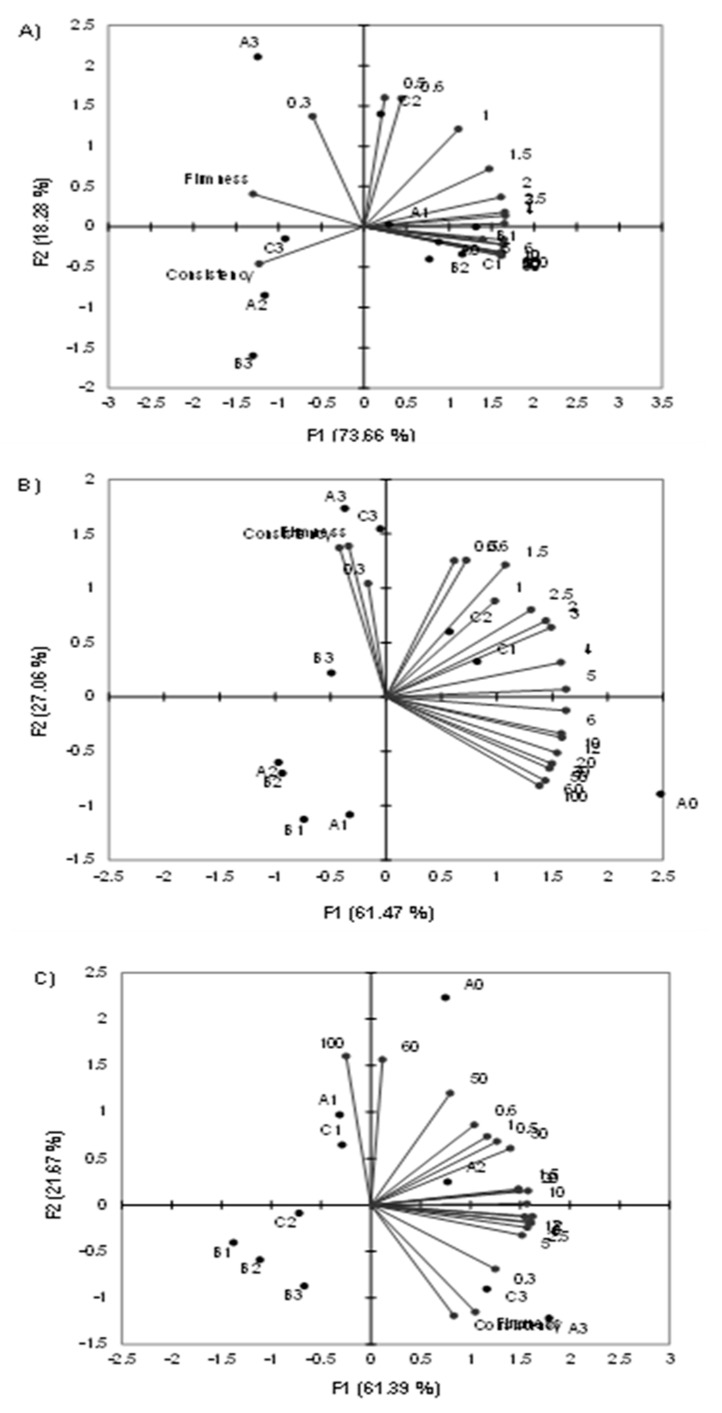
PCA (Principal Component Analysis) of flow and textural properties of pastry fillings at (**A**) 45 °C, (**B**) 60 °C and (**C**) 80 °C. The numbers on the charts represent shear rates (min^−1^).

**Table 1 foods-08-00066-t001:** Sample designations and cocoa, carob and LBG (locust bean gum) contents of the pastry fillings. The basic recipe comprised 45 g sucrose, 30 g vegetable fat and 10 g whole milk powder; these ingredients are not listed in the table.

SET (A)		SET (B)		SET (C)	
Sample Name	Composition	Sample Name	Composition	Sample Name	Composition
A0	15 g cocoa powder	B0	15 g cocoa powder	C0	15 g cocoa powder
A1	10 g cocoa powder 5 g seedless carob flour	B1	10 g cocoa powder 5 g carob flour with naturally present seeds	C1	10 g cocoa powder 5 g seedless carob flour 0.15 g LBG
A2	5 g cocoa powder 10 g seedless carob flour	B2	5 g cocoa powder 10 g carob flour with naturally present seeds	C2	5 g cocoa powder 10 g seedless carob flour 0.3 g LBG
A3	15 g seedless carob flour	B3	15 g carob flour with naturally present seeds	C3	15 g seedless carob flour 0.45 g LBG

**Table 2 foods-08-00066-t002:** Parameters of the Ostwald de Waele model estimated for cocoa and carob-based pastry mixtures. * marked values are significant at *p* < 0.05.

	45 °C					60 °C					80 °C				
Sample	*k* (Pas^n^)	*n*	*R* ^2^	RMSE (Pa)	MPE (%)	*k* (Pas^n^)	*n*	*R* ^2^	RMSE (Pa)	MPE (%)	*k* (Pas^n^)	*n*	*R* ^2^	RMSE (Pa)	MPE (%)
A0	4.58 *	0.74 *	0.9971	0.13	12.84	6.78 *	0.67 *	0.9999	0.02	4.77	3.94 *	0.53 *	0.9266	0.51	25.78
A1	4.08 *	0.81 *	0.9995	0.05	6.43	3.78 *	0.80 *	0.9993	0.05	9.24	3.55 *	0.67 *	0.9823	0.24	25.87
A2	2.53 *	0.78 *	0.9992	0.04	12.67	2.49 *	0.79 *	0.9986	0.05	7.13	2.72 *	0.39 *	0.8669	0.42	30.51
A3	2.24 *	0.76 *	0.9969	0.07	8.26	2.18 *	0.55 *	0.9653	0.20	19.13	2.61 *	0.20 *	0.9050	0.26	15.42
B0	4.58 *	0.74 *	0.9971	0.13	10.00	6.78 *	0.67 *	0.9999	0.02	4.77	3.94 *	0.53 *	0.9266	0.51	25.78
B1	5.15 *	0.78 *	0.9996	0.05	10.00	3.09 *	0.82 *	0.9996	0.03	5.13	2.37 *	0.82 *	0.9994	0.03	17.87
B2	4.63 *	0.78 *	0.9999	0.03	9.14	2.56 *	0.79 *	0.9987	0.05	8.98	2.42 *	0.74 *	0.9954	0.09	23.43
B3	2.90 *	0.82 *	0.9995	0.03	13.46	2.29 *	0.60 *	0.9906	0.11	20.97	2.32 *	0.53 *	0.9740	0.18	20.78
C0	4.58 *	0.74 *	0.9971	0.13	10.00	6.78 *	0.67 *	0.9999	0.02	4.77	3.94 *	0.53 *	0.9266	0.51	25.78
C1	4.84 *	0.76 *	0.9997	0.05	8.11	4.08 *	0.67 *	0.9874	0.23	11.58	3.25 *	0.59 *	0.9579	0.33	32.02
C2	3.49 *	0.76 *	0.9989	0.06	11.65	3.37 *	0.59 *	0.9830	0.21	10.76	2.88 *	0.73 *	0.9889	0.16	25.49
C3	2.77 *	0.81 *	0.9983	0.07	11.98	2.27 *	0.48 *	0.9651	0.19	24.29	2.08 *	0.14 *	0.8319	0.20	18.52

RMSE: root mean squared error; MPE: mean percentage error.

**Table 3 foods-08-00066-t003:** Parameters of the Herschel–Bulkley model estimated for cocoa and carob-based pastry mixtures. * marked values are significant at *p* < 0.05.

	45 °C						60 °C						80 °C					
Sample	*τ_y_*	*k* (Pas^n^)	*n*	*R* ^2^	MPE (%)	RMSE (Pa)	*τ_y_*	*k* (Pas^n^)	*n*	*R* ^2^	MPE (%)	RMSE (Pa)	*τ_y_*	*k* (Pas^n^)	*n*	*R* ^2^	RMSE (Pa)	MPE (%)
A0	−0.08	4.75 *	0.71 *	0.9969	9.69	0.15	0.07	6.62 *	0.69 *	0.9993	3.47	0.10	1.31 *	2.31	1.36	0.8807	0.63	21.80
A1	0.11 *	3.93 *	0.87 *	0.9997	7.12	0.04	0.09 *	3.63 *	0.88 *	0.9998	5.34	0.03	0.46 *	2.97 *	0.96 *	0.9984	0.07	8.48
A2	0.02	2.50 *	0.80 *	0.9993	7.83	0.04	0.13	2.32 *	0.89 *	0.9974	8.93	0.07	1.52 *	0.90 *	2.00 *	0.7917	0.46	27.48
A3	0.16 *	2.04 *	0.90 *	0.9976	9.62	0.6	0.41 *	1.70 *	0.95 *	0.9901	7.72	0.10	1.57 *	0.91	0.62	0.3754	0.83	21.33
B0	−0.08	4.75 *	0.71 *	0.9969	9.69	0.15	0.07	6.62 *	0.69 *	0.9993	3.47	0.10	1.31 *	2.31	1.36	0.8807	0.63	21.80
B1	0.04	5.09 *	0.79 *	0.9997	7.67	0.05	0.03	3.08 *	0.84 *	0.9992	6.59	0.05	0.07 *	2.28 *	0.89 *	0.9993	0.04	9.49
B2	0.05	4.53 *	0.80 *	0.9995	7.61	0.06	0.12 *	2.40 *	0.88 *	0.9986	7.64	0.05	0.17 *	2.20 *	0.89 *	0.9994	0.03	6.70
B3	−0.05	2.99 *	0.78 *	0.9991	6.40	0.05	0.25 *	2.00 *	0.80 *	0.9987	6.03	0.04	0.39 *	1.88 *	0.87 *	0.9950	0.08	8.43
C0	−0.08	4.75 *	0.71 *	0.9969	9.69	0.15	0.07	6.62 *	0.69 *	0.9993	3.47	0.10	1.31 *	2.31	1.36	0.8807	0.63	21.80
C1	0.02	4.82 *	0.76 *	0.9997	7.78	0.05	0.45 *	3.52 *	0.90 *	0.9981	7.22	0.09	0.57 *	2.54 *	1.02 *	0.9880	0.17	10.64
C2	0.19 *	3.25 *	0.86 *	0.9986	8.75	0.07	0.45 *	2.84 *	0.85 *	0.9973	8.55	0.09	0.34 *	2.45 *	0.98 *	0.9926	0.13	12.49
C3	0.15 *	2.59 *	0.91 *	0.9976	9.32	0.07	0.44 *	1.78 *	0.83 *	0.9919	7.66	0.09	1.45 *	0.93 *	1.59 *	0.8862	0.27	11.67

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
