# Peer review of "Influence of Carob Flour and Carob Bean Gum on Rheological Properties of Cocoa and Carob Pastry Fillings"

_foods, 2019, doi:10.3390/foods8020066_

Reviewer 1 Report

Results and discussion

Figure 2- a letter for significant difference is missing for sample A0.

Author Response

Reviewer 1 comment

Results and discussion

Figure 2- a letter for significant difference is missing for sample A0.

Response to Reviewer 1

Sorry for the missing letter. We have added the significance letter for sample A0 to figure 2.

Reviewer 2 Report

In this revised version, the authors have addressed some of the criticisms I raised in my review of the original manuscript.   However, one significant area that was not adequately addressed is the presentation of the modeling results.

The experimental data is noisy - this is understandable and expected.  However, the model fitting is presented with parameters to five significant figures.  The authors have not presented any sort of analysis of the uncertainty in the data to justify this level of precision.  Without this justification, presenting model parameters with five significant figures is misleading and not scientifically valid.

Author Response

Reviewer 2 comment

In this revised version, the authors have addressed some of the criticisms I raised in my review of the original manuscript. However, one significant area that was not adequately addressed is the presentation of the modelling results.

The experimental data is noisy - this is understandable and expected. However, the model fitting is presented with parameters to five significant figures.  The authors have not presented any sort of analysis of the uncertainty in the data to justify this level of precision. Without this justification, presenting model parameters with five significant figures is misleading and not scientifically valid.

Response to Reviewer 2

Dear Reviewer,

First of all, we the authors of the manuscript are more than grateful for your willing to improve our paper. We really appreciate your time and efforts. We have corrected the manuscript to show two significant figures. However, we must emphasize that we deliberately left 4 digits for the R2 and the RMSE values. Namely, if you look at the R2 and the RMSE values, it becomes obvious that presentation of data to only 2 significant figures is not enough because of the small order of magnitude and the nature of those parameters. In this case, their presentation to less than 4 digits results in possible oversight of some significant conclusions. Furthermore, the RMSE data were shown as the standard deviation of the residuals (difference between experimental and model predicted data), which is a measure of how far from the regression line those data points are, and is therefore used, by definition, as a mean od verification of experimental results, which also includes the data uncertainty.

The Authors have corrected the Tables according to Reviewers instructions, as well as the corresponding sections of the discussion and the conclusions. Thank you once again, for your efforts in order to improve our manuscript and future paper, we suppose.

Round  2

Reviewer 2 Report

My reservations about the modeling parts of this manuscript have not been alleviated.

Author Response

Dear Reviewer,

Considering your respected comments and suggestions for authors that (cit.) your “reservations about modelling parts of this manuscript have not been alleviated”, it is obvious that you’re, definitely, highly experienced expert for rheology. We, the authors of the paper are pleased that we get reviewer of your research profile and if is possible, we'll be glad to start a cooperation with you in the nearest future, if it would be possible. However, considering this manuscript, we have to emphasize that one of the basic principles in experimental research work is repeatability of the experiment. We, the authors of the manuscript are pretty sure that our experimental design is completely repeatable, thus every researcher can check the truthiness and reliability of the results exposed in this manuscript, by using the same methods explained in the manuscript within the chapter 2. “Materials and Methods”. We hope you agree with this remark, no matter how subjective it is, at the first glance.  We also have to point out that according to your recommendations considering the English language and style, we sent our manuscript to philologist and Anglicist who did some very minor corrections in the text, which are marked with ‘Strikethrough’ lines and the corrections are written with underlined red letters. Finally, in the name of my colleagues, I really wish to thank you for your time and efforts and please, do not hesitate to contact me/us if you’re interested for cooperation in future research.

Yours Sincerely, Siniša Srečec

Dr Siniša Srečec

Križevci College of Agriculture

Milislava Demerca 1, HR-48260 Križevci

CROATIA

tel. + 385 48 617 954

fax + 385 48 279 189

mobile: + 385 91 464 00 56

e-mail: ssrecec@vguk.hr

web: www.vguk.hr

This manuscript is a resubmission of an earlier submission. The following is a list of the peer review reports and author responses from that submission.

Round  1

Reviewer 1 Report

Unfortunately, this manuscript is not suitable for publication in its current form.  The major concern is that the authors have over-interpreted their data, and this calls into question the scientific validity of the conclusions drawn from these experiments.

Consider the rheological data presented in Figure 1.   It is not clear the degree to which the data is reproducible – how many trials were performed for each condition?   What are the reproducibility statistics and error bars on the graphs? The data for each experiment shows a lot of noise - are there really significant differences between the various curves in each panel? 

Notice also that the tree graphs for 60 C (Figures 1b, 1e, and 1h) all show a pronounced “step” at a shear rate of 10 min-1.  This signal cannot be real and suggests some problem with the experimental technique. 

Corresponding to the question about the precision of the experimental data, the numerical results shown in Tables 2 and 3 are not believable.  Values are reported with 4 and 5 significant figures – but given the noise in the experimental data, performing the modeling to this level of precision is simply not scientifically valid.  Without properly analyzing the data, it is not possible to draw any conclusions from the modeling exercise.  This same over-interpretation of the data pervades this manuscript.   Given the noise in the experimental data (and the fact that masses of the components used were only controlled to two significant figures), it would be difficult to believe that the results can be interpreted to more than two significant figures.

 There are a number of minor concerns as well:

1.  In Table 1, the components “45 g sucrose 30 g vegetable fat,  10 g whole milk powder” are repeated for each entry.  To save space, these common ingredients can be reported only once, and then the Table could contain only the information about the components that vary between sample sets and compositions.

2.  In Figure 1, each vertical axis is labeled as “viscosity” but the meaning of this is not clear since these are obviously shear thinning materials.   Is this actually the apparent viscosity (shear stress divided by shear rate)?   Or actual viscosity (the local derivative of the shear stress vs. shear rate curve)?   Was any adjustment made for a yield stress?

3.  Did the authors consider whether the samples may have violated the no-slip condition in either the rheological or textural studies?

Reviewer 2 Report

The manuscript deals with the influence of carob flour and carob bean gum on rheological properties of cocoa and carob pastry fillings.

 Abstract

This section must be improved. Please present your main results, i.e. values.

Materials and methods

Texture analysis, adhesiveness??

Table 1- Please align the text and columns width.

Line 83- “Samples were mixed in a kitchen blender (Philips, Amsterdam, Netherlands) for 7 minutes to obtain a homogenous blend.”?? Type of mixer/turbine used??

 Results and discussion

Pictures of each sample??

Texture analysis, adhesiveness??

Figure 1 has low quality and must be improved.

Figure 2, letters for significant differences??